# Biosensors for the Detection of Bacterial and Viral Clinical Pathogens

**DOI:** 10.3390/s20236926

**Published:** 2020-12-04

**Authors:** Luis Castillo-Henríquez, Mariana Brenes-Acuña, Arianna Castro-Rojas, Rolando Cordero-Salmerón, Mary Lopretti-Correa, José Roberto Vega-Baudrit

**Affiliations:** 1National Center for High Technology (CeNAT), National Laboratory of Nanotechnology (LANOTEC), San José 1174-1200, Costa Rica; luis.castillohenriquez@ucr.ac.cr; 2Physical Chemistry Laboratory, Faculty of Pharmacy, University of Costa Rica, San José 11501-2060, Costa Rica; 3Chemistry School, National University of Costa Rica, Heredia 86-3000, Costa Rica; mbrenesacua@gmail.com (M.B.-A.); ariannac.r9vi@gmail.com (A.C.-R.); rocordero105@gmail.com (R.C.-S.); 4Nuclear Research Center, Faculty of Science, Universidad de la República (UdelaR), Montevideo 11300, Uruguay; mary@cin.edu.uy

**Keywords:** bacterial detection, biosensors, clinical pathogen, COVID-19, electrospun nanofibers, nano-biosensors, point-of-care, SARS-CoV-2, viral detection

## Abstract

Biosensors are measurement devices that can sense several biomolecules, and are widely used for the detection of relevant clinical pathogens such as bacteria and viruses, showing outstanding results. Because of the latent existing risk of facing another pandemic like the one we are living through due to COVID-19, researchers are constantly looking forward to developing new technologies for diagnosis and treatment of infections caused by different bacteria and viruses. Regarding that, nanotechnology has improved biosensors’ design and performance through the development of materials and nanoparticles that enhance their affinity, selectivity, and efficacy in detecting these pathogens, such as employing nanoparticles, graphene quantum dots, and electrospun nanofibers. Therefore, this work aims to present a comprehensive review that exposes how biosensors work in terms of bacterial and viral detection, and the nanotechnological features that are contributing to achieving a faster yet still efficient COVID-19 diagnosis at the point-of-care.

## 1. Introduction

Biosensor’s concept was firstly addressed by Clark and Lyons around 1962 when they developed an oxidase enzyme electrode for glucose detection [1]. Since then, nanotechnological development has promoted biosensors evolution and specialization for different purposes [2]. Currently, nanotechnology is at the forefront of science, and its combination with biosensoring applications involves different fields such as medicine, biology, environmental, drug delivery, and food safety [3,4,5,6,7]. However, the detection of pathogens has become one of the most relevant objectives for these devices since bacterial and viral diseases currently represent an important thread for human health [8,9].

Virus and bacteria detection commonly involves the use of several molecular techniques such as the reverse transcription-polymerase chain reaction (RT-PCR), which remains the gold standard for pathogen detection [10]. The classical detection methods for these pathogens usually require isolation, culturing and, biochemical tests [11]. Additionally, serological tests like the Enzyme-Linked Immunosorbent Assay (ELISA) are used for the detection of antibodies and immunoglobulin needed for identification purposes [12]. However, some of these techniques take a long time to obtain results and are usually laborious. Therefore, new approaches based on nanotechnological advances have emerged as suitable and easier options for detecting pathogens in faster and efficient ways [11,13].

On one hand, nanoparticles (NPs) have demonstrated outstanding properties against different pathogens used to develop novel devices and technologies that contribute to this public health issue [14,15]. The interest is not limited to human diseases, but also considers the ones affecting animals since zoonosis is an existent thread. Stringer et al. developed an optical biosensor using gold NPs (AuNPs) and quantum dots (QDs) for the detection of porcine reproductive and respiratory syndrome virus [16].

On the other hand, the international scientific community’s interest in using DNA biosensors or sequence-specific DNA detectors for clinical studies is increasingly growing. In 2007, Dell’Atti et al. developed a combined DNA-based piezoelectric biosensor for simultaneous detection and genotyping of high-risk human papilloma virus (HPV) strains [17]. In addition, these biosensors have been employed for DNA damage research and specific gene sequences detection [18,19].

Biosensors and nano-biosensors have been extensively used for the detection of viral and bacterial clinical pathogens. These devices are practical (e.g., enable point-of-care (POC) testing through smartphone-based nano-biosensor), fast, and are considered as innovative technologies that provide an alternative solution to the mentioned disadvantages presented by common detection methods [20,21,22]. These technologies have been employed for studying viruses affecting human health such as Ebola virus, human immunodeficiency virus (HIV), and more recently the newly discovered acute respiratory syndrome coronavirus 2 (SARS-CoV-2), as well as bacteria like *Escherichia coli* and *Salmonella* spp. [23,24,25,26,27].

The literature search for this review was conducted in Science Direct, PubMed, Scopus, and Web of Science databases to identify studies in the fields of nanotechnology, nano-biotechnology, and electronics that reported the use of biosensor technologies for bacterial and viral pathogen detection in the title and/or abstract. Here we present a comprehensive and integrative update of the topic based on the main findings of 223 papers published between 2010 and 2020. Therefore, this review aims to expose how biosensors work in terms of bacterial and viral detection, describing the nanotechnological features such as NPs, graphene QDs (GQDs), and electrospun nanofibers, which enhance their affinity, selectivity, and efficacy in detecting these pathogens, as well as highlighting current advances for the COVID-19 pandemic assessment at the POC.

## 2. Biosensors

Biosensors can be defined as a measurement system for analyte detection that combines a biological component with a physicochemical detector [28]. The analyte detection depends on the biosensor design and purpose. Some commonly used devices such as smartphones can be employed as a biosensor with the inclusion of simple accessories as published by Soni et al., where they developed a non-invasive smartphone-based biosensor for urea using saliva as sample [29,30]. This allows fast and low-cost preliminary detection [31].

Usually, biosensors detect biomolecules such as nucleic acids, proteins, and cells that are associated with diseases. This is possible because of their three major components: The biologically sensitive element, the detector element, and the reader device [32]. Enzymes, microorganisms, organelles, antibodies, and nucleic acids are used to detect the biomolecules [33]. In addition, researchers must identify the requirements to obtain a functional device according to the intended use. Hence, multidisciplinary studies are fundamental to select the proper material, transducing device, and biological element involved before assembling the biosensor [34].

At a clinical level, biosensors are applied for detecting disease-associated biomolecules [32]. These devices can monitor the biochemical markers of a disease in body fluids, such as saliva, blood, or urine [35,36]. Zhang et al. developed a non-invasive method for glucose testing based on a disposable saliva nano-biosensor to improve patient compliance, reduce complications, and costs derived from diabetes management. In the clinical trials, they obtained outstanding results in terms of accuracy compared to the UV spectrophotometer. Thus, the disposable device can be presented as an alternative for real-time salivary glucose tracking [37].

Biosensors can be applied for many other clinical diagnostic purposes, such as cholesterol, markers related to cardiovascular diseases, biomarkers of cancer or tumors, allergic responses, disease-causing bacteria, viruses, and fungi infections [38,39,40,41]. Aside from that, biosensors can be employed for bacteria and virus detection in food and water, which are potential sources of diseases [42,43]. Zhao et al. fabricated a low-cost, portable microfluidic chemiresistive biosensor based on monolayer graphene, AuNPs, and streptavidin-antibody system for the rapid in-situ detection of *E. coli*. In this case, the bacteria are captured on the biosensor’s surface and detection is performed through electric readouts [44]. Another approach published by Samanman et al. describes the development of a glutathione-S-transferase tag for white spot binding protein (GST-WBP) immobilized onto a gold electrode through a self-assembled monolayer. This biosensor can detect white spot syndrome virus (WSSV) in shrimp pond water due to binding between WSSV and the immobilized GST-WBP [45].

### 2.1. Operating Principles

Biosensors are constituted by three components (Figure 1) [38,46]. These devices have sensing elements, also called bioreceptor that emulates in vivo molecular recognition phenomena [47]. There is a wide range of sensing elements such as cells, microbes, cell receptors, antibodies, enzymes, or nucleic acids [48,49,50,51,52]. These biological sensitive elements recognize the analyte and interact with it depending on the type of biosensor [53]. One of the main biorecognition strategies is based on bacterial or viral nucleic acid sequences [54,55]. Solanki et al. developed a DNA bioelectrode to detect *Vibrio cholerae*, which is stable for at least 15 weeks under 4 °C storage. The biosensor consisted of O1 gene-based 24-mer single-stranded DNA probe immobilized onto sol-gel derived nanostructured zirconium oxide (NanoZrO_2_) film [56].

The second element is the transductor or detector, which works by sensing the signal related to a physicochemical change caused by the interaction between the bioreceptor and the analyte. It transforms the signal into another one that can be evaluated and quantified [57,58,59,60,61]. The last part of a biosensor is the reader device. It usually involves a display that depends on software and hardware to generate the results [62].

Some important attributes define the performance of a biosensor. In the first place, selectivity is the capacity of a bioreceptor to detect a specific bio-entity when analyzing a sample composed of other components. This is probably the main feature and determines the needed bioreceptor. Second, reproducibility is the ability to produce the same response for a certain experimental set-up that is performed multiple times. Reproducible signals provide high reliability and robustness. Third, stability is the capacity to endure ambient disturbances around the system that can affect the precision and accuracy of the device. Fourth, sensitivity also known as the limit of detection (LOD) is the minimum amount of the analyte that can be detected by a biosensor. For clinical applications, it is required to detect the analyte in samples of low concentrations (ng/mL or fg/ml). Finally, linearity examines how accurate are the measurements within the analyte range of concentrations (i.e., linear range), and in response to the smallest variation in concentration that can cause a change in the output (i.e., resolution) [63].

### 2.2. Types of Biosensors

Biosensors can be classified by the way they transduce signals into optical, electrochemical, and piezoelectric devices [57,58,59,60,61,64]. Optical biosensors are those that perform their analysis through the measure of photons, using optic fibers as transduction elements [58,59,65]. Several optic sensing mechanisms can be employed by this type of biosensor for analyte detection such as absorption, colorimetry, fluorescence, or luminescence [66]. This kind of biosensor presents a lower noise and immunity to electromagnetic interference, which gives it an advantage over electrochemical and piezoelectric biosensors [67].

Vidal et al. developed a chromatic biosensor for quick bacterial detection based on polyvinyl butyrate-polydiacetylene non-woven fiber composites. The device shows promising potential to alert about possible infections caused by *Staphylococcus aureus*, *Micrococcus luteus*, and *E. coli* [68]. In another study, Jeong et al. constructed a fluorescent supramolecular biosensor for bacterial detection. The binding of these pathogens induces conformational changes in the supramolecular state, which causes a fluorescence emission that can selectively detect *E. coli* over other microorganisms [69]. Regarding viral analysis, Ahmadi et al. evaluated single virus detection through an optical biosensor, where viral particles attached to a microsphere optical resonator’s surface caused a shift of resonance to longer wavelengths [70].

Furthermore, Surface Plasmon Resonance (SPR) is an optical technique that has contributed greatly to immunoassays development. This type of resonance occur when a methalic thin-film is deposited on a dielectric waveguide, where the intensity data from the reflection of *p*-polarized light (i.e., along the plane of incidence) is used. On the other hand, SPR enhanced ellipsometry, also called Total Internal Reflection Ellipsometry (TIRE), uses the reflection properties of *s*-polarization (i.e., perpendicular to the plane of incidence) [71,72]. SPR-based biosensors can perform simultaneously the detection of multiple biomolecules and real-time monitoring interactions of chemical and biological analytes such as RNA, DNA, ligands, and cofactors, with label-based or label-free form [73]. In addition to that, these biosensors provide many other advantages in quantifying low molecular weight analytes, rapid detection, low-cost, and high reliability, specificity, and reproducibility that make them suitable for clinical applications [74].

The second type, electrochemical biosensor, has been extensively applied to pathogen detection. These devices sense the analyte through electrodes by measuring electrical signals resulting from catalytic reactions or specific unions. The previous is derived from the capture of electrons as a result of redox reactions between the analyte and the bio-element [75]. In addition to that, the analysis of the desired element is determined by different readouts like potentiometry, amperometry, and conductometry [76]. This type of biosensor has been subjected to improvements due to bio- and nanomaterials development [76,77].

Recently, Mathelié et al. employed non-cytotoxic silica NPs-assisted electrochemical biosensor for sensitive and specific detection of *E. coli*. The electrochemical immune-biosensor detects the bacteria in five minutes by cyclic voltammetry measurements, and also represents a potential device for targeting a variety of other microorganisms through little modifications within its features [78]. In another study, Baek et al. developed an electrochemical biosensor composed of eight novel peptides separately in a gold electrode for the detection of human norovirus. The peptides exhibited a high binding affinity towards the viruses, and a decrease in current signals explained by increasing concentration of the virus [79].

Finally, yet importantly, there are piezoelectric biosensors. Piezoelectricity refers to the ability of a material to generate a voltage under mechanical stress [80]. These biosensors possess crystals that vibrate under the influence of an electric field. Besides, certain materials vibrate at characteristic resonant frequencies in response to interaction with other molecules. The relationship between the resonant frequency changes and the mass from the molecules adsorbed or desorbed from the crystal’s surface is conceived as the working principle of transduction in this type of biosensor. Therefore, vibration provides information on the phenomenon that is being measured [81,82].

Fu et al. discuss the advances in piezoelectric thin films acoustic wave devices for bacterial and viral detection of pathogens adsorbed on surfaces through DNA interaction with complementary strands. The previous allows early detection of clinical pathogens, and thus, prevents the spreading of the infection [83]. In another approach, Guo et al. worked on sensitive *E. coli* O157:H7 detection system using a piezoelectric biosensor-quartz crystal microbalance with antibody-functionalized AuNPs to enhance changes in detection signals. It was demonstrated that the developed device can be used as a suitable real-time monitoring method for the mentioned pathogen [84].

## 3. Biosensors Nanotechnological Features for Bacterial and Viral Detection

Over time, many techniques and methods have been developed for detecting pathogens such as viruses and bacteria, including colorimetric methods, fluorescence polarization, and electrochemical analysis [85]. However, those are very expensive and possess limitations related to time-consumption, low precision of the results, poor stability, and short life span [86].

Bacterial and viral outbreaks have caused many issues in biomedical, food, and environmental context, making necessary the development of new strategies that allow faster detection of these pathogens to effectively contain and control their impact on human health [87]. The combination of nanotechnologies and biosensors’ characteristics is currently being considered as a potential opportunity for speeding up the development of fast, highly sensitive, and specific devices for genuine bacterial and viral detection. As a consequence, nano-biosensors make use of chemical, electrical, optical, and magnetic properties of materials for detecting biomolecules and pathogens [88,89].

In order to satisfy the previous, nanotechnology has greatly contributed to the development of biosensors due to research in nanomaterials and nanostructures, such as carbon nanotubes, GQDs, metal oxide NPs, metal nanoclusters, plasmonic nanomaterials, polymer nanocomposites, nanogels, among others (Figure 2) [90,91,92,93]. These have been employed for modifying electrode surfaces to improve critical features, such as reproducibility, selectivity, and sensitivity, due to their biocompatible character, structural compatibility, and high adsorption capacity. Therefore, nanomaterials have demonstrated to be suitable for biosensing applications, enhancing the performance with increased sensitivities and lower detection limits [94].

Additionally, different nanomaterials have been used to increase the immobilized bioreceptor loadings. However, the strategy for immobilizing the bio-specific entity onto the nanomaterial is considered the biggest challenge for developing a high quality and reliable nano-biosensor. Non-covalent approaches such as electrostatic interactions, polymers entrapment, or van der Waals forces between the nanomaterial and the biomolecule do not alter their specific properties. On the other hand, covalent binding provides more stability and reproducibility of surface functionalization, as well as reducing the risk of unspecific physisorption. Although the previous techniques represent good strategies for binding biological species to surfaces, supramolecular interactions have recently been considered as superior since these are reversible, enabling the regeneration of the transducer element [95,96].

Regarding other uses, nanomaterials can perform as nanocarriers for signaling elements, as well as signal amplification. Depending on the chemical composition, nanomaterials can be subject to direct functionalization during synthesis, or functionalized by coating using functional polymers [97]. Nanomaterials functionalization provides three important advantages: reproducible immobilization of bioreceptor units, increase the biocompatibility, and the development of label-free transduction techniques [96].

Moreover, nano-biosensor materials’ high surface area is considered a major advantage compared to conventional devices, and plays an important role in the sensitivity and fast response of the devices [98,99]. Therefore, these are conceived as excellent tools used for the detection, function, and interaction of proteins and nucleic acids, which improve the quality and performance of diagnosis for bacterial and viral diseases [100]. The following sections present an overview of some promising nanotechnological features in biosensors.

### 3.1. Nanoparticles

NPs are a wide range of materials with dimensions below 100 nm that have been used in various areas such as medical, pharmaceutical, manufacturing and materials, environmental, electronics, and mechanical industries due to their multiple properties [101,102,103,104]. Among the mostly employed are metal NPs such as AuNPs and silver NPs (AgNPs), which can be produced in different sizes and shapes (e.g., nanospheres, nanocylinders, nanowires, and nanocages). These NPs exhibit low toxicity, as well as multiple interesting chemical, biological, and physical properties, such as photo-thermal, optical, electrochemical, and biocompatibility based on their inert nature in biological fluids [105,106,107]. Additionally, these NPs can be synthesized with ease, fulfilling relevant roles for diagnostic probes, and functionalized due to the presence of functional groups for achieving ligand-binding functions with a wide range of molecules, such as antibodies or genetic material [108,109].

An important application of nano-biosensors composed of metal NPs is related to waterborne diseases. In these cases the infection is usually linked to microbial contamination due to several pathogens, including bacteria. Nanotechnological detection systems with optical sensing have been used for these pathogens [110]. Elahi et al. designed a highly sensitive fluorescence nano-biosensor for the detection of *Shigella* species. To achieve a satisfactory design, two DNA probes as sensing elements were immobilized on the surface of AuNPs synthesized for the development, forming a DNA-probe AuNPs-fluorescence system. The research group also synthesized iron NPs (MNPs) that were later modified with Sulfosuccinimidyl 4-Nmaleimidomethyl cyclohexane-1- carboxylate (SMCC). A second system constituted by a third DNA probe immobilized on MNPs was formed for separating target DNA. The results exhibited an increasing fluorescence intensity with an increase of target DNA concentration [111].

In another study, carried out by Takemura et al., an ultrasensitive, rapid, and specific localized SPR-induced immunofluorescence nano-biosensor was developed for detecting influenza virus. Researchers employed AuNPs-induced QD fluorescence signal conjugated with antineuraminidase antibody (Anti-NA Ab) and conjugation of anti-hemaglutinin antibody (anti-HA Ab) to the QDs. The device successfully detected influenza virus H1N1. However, due to its versatility, it was also possible to detect clinically isolated influenza virus H3N2 and norovirus-like particles [112].

### 3.2. Graphene Quantum Dots

GQDs are among the most fascinating carbon-based nanomaterials employed for the development of biosensors, mostly electrochemical. These materials present outstanding properties such as signal amplifying characteristics, biocompatibility, tunable size, electro-catalytic performance, and capacity to detect multiple biomolecules. Additionally, their inertness, non-toxicity, long-term chemical stability, and water stability make them very valuable for biomedical applications [94,97].

GQDs obtained through different synthesis methods have been used for biosensing applications since their large surface area can be functionalized. This allows them to directly detect DNA, enzymes, proteins, antigens, antibodies, and other biomolecules by the oxide components formed on their surface during the synthesis process [113]. Safardoust et al. synthesized GQDs from citric acid and ethylene diamine to use them as a photoluminescence sensor for detecting *S. aureus* and *E. coli*. This biosensor demonstrated a linear relationship between the fluorescence intensity and the concentrations of the bacteria up to 9 × 10^7^ CFU/mL [114].

In another approach, Hazani et al. fabricated a highly sensitive electrochemical peptide nucleic acid (PNA) biosensor based on functionalized graphene oxide composited with cadmium sulfide QDs (CdS QDs). The device was developed for detecting *Mycobacterium tuberculosis* and showed a LOD of 8.948 × 10^−13^ M [115]. Furthermore, GQDs integration into a biosensor can improve its performance in terms of reproducibility, selectivity, and sensitivity [116].

### 3.3. Electrospun Nanofibers

Electrospinning is a nanotechnological method in which an electrostatic field force applied to a polymer solution causes a charged liquid jet to moves downfield towards an oppositely charged collector, where fine fibers are deposited [117]. Electrospun nanofibers have been the target of different applications like drug delivery systems or scaffolds for skin tissue engineering due to their structure and physicochemical properties, such as a large surface area to volume ratio, small particle size, and high porosity [117,118,119,120]. However, a novel application is their use for developing nano-biosensors focused on detecting viral and bacterial pathogens [121,122,123,124].

Nano-biosensors development using these nanostructures can be achieved by two approaches. On one hand, functional polymers are electrospun to obtain a nanofiber that is used directly as an inducing element of the corresponding biosensor, which will present fast response time, high sensitivity, and good biocompatibility. On the other hand, electrospun nanofibers are used as templates to which a sensitive material is deposited on their surface, and later the system is subjected to chemical modification to produce a composite film on an electrode, with nanostructures that have the intended sensing characteristics [125,126].

Although the manufacturing process is simpler, keeping bio-receptor functionality is considered a great challenge for the production of this type of device. The sensing element can be immobilized through different strategies according to its physicochemical characteristics, as well as the ones from the nanofiber scaffolds, and also, based on their interfacial interactions [127].

Moreover, this type of nano-biosensor is based on various sensing principles such as optics, electric resistance, photoelectricity, vibration frequency, electric current, and others [128,129,130,131,132]. Luo et al. developed a nitrocellulose electrospun nanofibrous capture membrane for detecting *E. coli* O157:H7 and bovine viral diarrhea virus. The device’s design was based on capillary separation, and conductometric immunoassay using a silver electrode. Nanofiber antibody’s surface functionalization and sensor assembly process allowed retaining the unique fiber morphology, and displaying a linear response to both pathogens with a detection time of 8 min [133].

Quiros et al. prepared electrospun membranes composed of polyacrylonitrile (PAN) and poly(4-vinylphenylboronic acid-*co*-2-(dimethylamino)ethyl methacrylate-*co*-n-butyl methacrylate)(pVDB) for fast sensing of bacteria. The pVDB@PAN membranes were used as fluorescent bacterial biosensors, displaying maximum fluorescence intensity after 24 h in contact with *S. aureus* or *E. coli*. Meanwhile, the membranes became non-responsive within 8 h in contact with *Pseudomonas putida* due to the rapid formation of bacterial biofilm that blocked the membrane surface, disrupting fluorescence readings. This development can be useful for the early identification of pathogenic bacteria as an attempt to prevent their spreading [134].

Some research groups have designed nano-biosensors based on electrospun nanofibers for viral detection as well. Tripathy et al. worked on an ultrasensitive electrochemical platform with electrospun semi-conducting Manganese (III) Oxide (Mn_2_O_3_) nanofibers for detecting DNA hybridization. This biosensor makes use of electrochemical transduction techniques for zeptomolar (i.e., 10^−21^ M) detection of Dengue primer, resulting in a limit of detection of 120 × 10^–21^ M [135].

Therefore, nanofiber-based biosensors present advantages over the conventional ones such as polymer diversity for its manufacture, high specific surface area with high responsiveness, as well as an outstanding sensibility [136,137,138].

## 4. Bacterial and Viral Pathogens Detected through Biosensors and Nano-Biosensors

Conventional clinical analyses including an antibody or nucleic acid-based, biochemical, and enzymatic methods, are very reliable but take a long time to obtain a result. Health disciplines demand the acquisition of faster outcomes to speed up the appropriate treatment [139,140]. In this sense, biosensors and nano-biosensors are useful tools that offer an accurate response in shorter times due to their ability to provide real-time and faster clinical results [141]. Currently, there is an increasing interest in their use to detect pathogens in the human body (Table 1) [140].

Molecular determination demands to improve the analytical performance of biosensors, which have enhanced their unique features to develop POC devices in order to run a rapid and cost-effective analysis of complex biological matrices [156]. Commercial versions of these devices are available to detect pathogens such as *E. coli*, *Helicobacter pylori*, influenza A and B viruses, HIV, tuberculosis, and malaria [157]. Advantages such as small samples and low energy required to avoid complications in terms of transportation and processing, make them suitable for easy and fast use in the identification of bacterial and viral pathogens [141]. Needless to say, nanomaterials advances have benefited biosensor performance to achieve the task [158].

### 4.1. Bacterial Pathogen Detection

Focusing on the human body, bacterial infections caused mainly by Gram-negative microorganisms represent a particular challenge in human health worldwide. Multidrug resistance variants have been greatly influenced by their indiscriminate exposure to antibiotics discharged in water, addition to food or more commonly, due to improper use of these drugs from patients [159].

Since the previously mentioned is considered a major current health concern, different kinds of nanomaterials and biorecognition elements have been employed to develop biosensors for antibiotic detection, as well for bacteria [160]. Common pathogenic bacteria include *E. coli*, *Salmonella typhi*, *Clostridium perfringens*, and *Shigella* spp., which can cause different kinds of diseases in humans, animals, and plants [161]. However, *S. aureus* is recognized as one of the most fatal bacteria that can cause rapid mortal infections and is often resistant to multiple antibacterial active substances. Thus, it is necessary to develop new approaches for easier and faster detection since conventional culture methods require 3–5 days to obtain results, and other nucleic acid-based methods are expensive and imply trained personnel [162,163].

Suaifan et al. developed a biosensor able to detect *S. aureus* in a few minutes. The sensing tool is based on the proteolytic activity of the pathogen proteases on a specific peptide substrate placed in the middle of two magnetic nanobeads. In this case, the dissociation of magnetic nanobeads-peptide moieties results in color change [164]. In another approach, Ahari et al. constructed a potentiometric nano-biosensor able to detect the bacteria through the identification of an exotoxin emitted by the microorganism. Particularly, the method is often used for contaminated food, but it can also be applied for clinical detection [165].

Starodub et al. designed high-specific biosensors based on SPR and TIRE for *Salmonella* spp. detection. These devices employed Ag-Ab reactions and a surface binding layer as the reactive part. A sensitivity within 10^1^–10^6^ cells/mL was reported for the SPR biosensor, while the TIRE exhibited a higher sensitivity up to several cells in 10 ml [166]. In another study, Vaisocherová et al. used a SPR biosensor based on ultra-low fouling and poly(carboxibetaine acrylamide) for the detection of the same bacteria in food. They reported a LOD of 7.4 × 10^3^ CFU/mL and average response time of 80 min. Additionally, these biosensor was found to be useful for the simultaneous detection of *E. coli*. [167].

Another important bacteria, *V. cholerae*, is a Gram-negative facultative anaerobe that causes Cholera disease. People would infect by consuming contaminated liquids or food, providing an ideal platform for the disease, which also spreads quickly due to its secretory nature. Therefore, its diagnosis plays an important role in the disease assessment because of its mortal rate rounds between 50–60% [168]. Recently, Narmani et al. developed an ultrasensitive and selective fluorescence DNA biosensor based on AuNPs and magnetic NPs for the determination of the bacteria’s O1 OmpW gene [169].

The Gram-negative bacteria, *Shigella,* belongs to the *Enterobacteriaceae* family. Infected people develop diverse symptoms including diarrhea, cramps, fever, and vomit. According to the World Health Organization (WHO), the annual number of *Shigella* cases worldwide is approximately 164.7 million with 1.1 million of those resulting in death, and the majority of them involve young children under the age of 5 years old [170]. Research performed by Elahi et al. discovered an early detection method of infectious *Shigella*. In this study, AuNPs-DNA probes were hybridized with Spa gene sequence in order to create an optical genosensing system. This biosensor makes the sample solution turns to purple in the absence of the complementary target, whereas the solution remains red in the presence of the specific gene sequence [171]. On the other hand, Xiao et al. covalently immobilized a DNA probe onto fiber-optic biosensors able to hybridize with a fluorescently labeled complementary DNA. The obtained results were comparable to the ones obtained by PCR, which suggests considering this method as an alternative for *Shigella* detection [170].

The different approaches for biosensoring detection of pathogenic bacteria have been successful and are currently being considered by many health governments and research institutions, mainly because of their fast response, high-quality performance, and reliable results [172,173,174].

### 4.2. Viral Pathogen Detection

Viral pathogen diagnosis is important for early and effective treatment of patients in order to prevent outbreaks or pandemics. For that reason, biosensors are being widely employed for making diagnosis easier, avoiding hard proteins or DNA identification techniques in specific virus [175,176]. One of the most common and dangerous viral pathogens is the influenza virus because of its ability to spread easily and constantly mutation. Hence, detection at early stages can be difficult [177,178].

Hassanpour et al. developed a novel optical biosensor composed of pDNA bioconjugated citrate capped AgNPs towards target sequences for ultrasensitive and selective *Haemophilus influenza* detection in human biofluids [179]. This pathogen has also been detected through other different biosensors, including the work reported by Jiang et al. [151,179,180,181]. The paper describes the development of a polydiacetylene sensitive biosensor using antibody detection for H5N1 (avian influenza), in which the polydiacetylenes vesicles show a dramatic change in color from blue to red upon the detection of the virus [181]. In another approach, Lee et al. also fabricated a label free localized SPR biosensor for the detection of H5N1 with a LOD of 1 Pm (i.e., 10^−12^ M). However, the device was constructed with a multifunctional DNA 3 way-Junction immobilized onto a hollow Au spike-like NP. The bioprobe demonstrated an adequate target recognition and the capacity to provide signal amplification [182].

Other dangerous viruses that affect the population worldwide include ebolavirus, HIV, and Hantavirus [183,184,185]. The first one is a negative strand-RNA virus that belongs to the *Filoviridae* family and causes a deadly disease called Ebola. The infected people with this agent develop a series of symptoms, where hemorrhagic fever is considered as fatal [186,187,188]. Currently, there is no vaccine or specific treatment [189]. However, different studies have presented the development of biosensors for detecting this pathogen [190]. Ilkhani et al. fabricated a novel electrochemical-based-DNA biosensor through enzyme-amplified detection to improve the sensitivity and selectivity of the device for the pathogen [191]. In addition to that, Baca et al. developed a biosensor that can detect the virus within 10 min at the POC by using surface acoustic waves, showing potential to detect it before symptoms onset [192].

On the other hand, HIV is a retrovirus that attacks a patient’s immune system, causing an inability to resist many diseases, and culminating in death when the person is not under drug control. Clinical treatments for HIV are crucial for reducing mortality, but early diagnosis saves many lives as well and can decrease spread rates [193,194,195]. Shafiee et al. worked on a photonic crystal biosensor to detect multiple HIV-1 subtypes (A, B, and D) upon binding of the biological analyte with the biosensor [196]. In addition to that, Gong et al. prepared a nanocomposite of polyaniline/graphene (PAN/GN) using reverse-phase polymerization for the development of an electrical DNA-biosensor that showed great selectivity, and sensitivity for the detection of HIV-1 gene fragment [197].

Hantavirus is a cluster of viruses that are part of the *Bunyaviridae* family. The spread begins through contact with liquids, food, or particles contaminated with rodent excreta. It causes hemorrhagic fever, respiratory insufficiency, and heart failure within 2–7 days after getting infected [198,199]. Regarding its detection, Gogola et al. have performed important research for the development of biosensors [200,201]. In a first approach, they prepared an electrochemical immunosensor based on chemical modification of the gold surface with the virus antigen/protein [200]. In a second study, the research group designed a quick electrochemical biosensor based on biochar (BC) as a carbonaceous platform for immunoassay applications due to its highly functionalized surface for covalent binding with biomolecules [201]. Both studies developed devices as promising and suitable tools for hantavirus clinical detection [200,201].

Furthermore, several bio-elements can be incorporated into a biosensor for virus detection including markers, RNA, structural proteins, and enzymes from the viral pathogens [202].

#### COVID-19 Pandemic

Currently, many viruses are being considered to have the capacity of causing future pandemics. Different factors such as fast dissemination, a high transmission rate of new variants, difficulties to develop efficient and sensible diagnostic techniques, as well as the lack of specific vaccines and safe drugs for treatment, make them one of the major threats for mankind [203,204]. The most recent case is the COVID-19 announced as a pandemic on March 13th, which is an infectious disease with rapid human-to-human transmission caused by SARS-CoV-2. This pathogen belongs to the positive-strand RNA viruses [205,206].

Like any other viral outbreak, an early diagnosis is fundamental for preventing an uncontrollable spread of the disease. However, this pandemic has the particularity that more than 30% of the confirmed cases are asymptomatic, thus making it harder to control [206,207,208]. RT-PCR is the most used suitable and reliable method for detecting SARS-CoV-2 infections until now. Nevertheless, the technique is time-consuming, labor-intensive, and unavailable in remote settings [209,210]. Although several other methods can be employed for that purpose, such as immunological assays, thoracic imaging, portable X-rays, or amplification techniques, the pandemic spread of COVID-19 demands to develop POC devices for rapid detection (Figure 3) [211,212,213,214].

Sheridan states that there are two types of rapid POC biosensors for COVID-19 detection. In the first place, there is a nucleic acid test, which consists of detecting the virus in the patient’s sputum, saliva, or nasal secretions [215,216]. The other type commonly employed is the antibody test that is done through the analysis of collected blood samples five days after the initial infection, which is when the immune response causes the production of IgM and IgG due to the presence of the virus [217,218,219].

The industrial sector has developed some suitable POC biosensors for the qualitative detection of SARS-CoV-2 IgM and IgG antibodies using samples as low as 10 µL of human serum, whole blood, or finger prick, obtaining results within 10–15 min (Table 2) [220]. Many of these rapid serological tests are paper-based biosensors that perform a colorimetric lateral flow immunoassay. In this method, SARS-CoV-2 specific antigens are typically labeled with gold, and bind the corresponding host antibodies, which migrate across an adhesive pad. As can be seen in Figure 4, anti-SARS-CoV-2 IgM antibodies interact with fixed anti-IgM secondary antibodies on the M line, while IgG antibodies interact with anti-IgG antibodies on the G line. Therefore, M or G lines only appear if the sample contains SARS-CoV-2 specific antibodies, otherwise, only the control line (C) will be shown [221]. Although the use of serological tests to detect SARS-CoV-2 is still under debate, these are foreseeing as crucial tools for the implementation or ceasing of lockdowns established worldwide [222].

Other research groups have developed Lab-on-a-Chip-based biosensors for SARS-CoV-2 detection [214,223]. This technology avoids the need for specialized personnel through the integration of microfluidic components into a biosensor, allowing increasing their production, and reducing the costs of the assay [224]. POC commercialized instruments based on this microfluidic technology are having an important role in this pandemic, like ID NOW^®^, Filmarray^®^, GeneXpert^®^, and RTisochip^®^ [225].

Cell-based biosensors have also contributed to COVID-19 diagnosis. Mavrikou et al. developed a biosensor based on membrane-engineered mammalian cells that possess the human chimeric spike S1 antibody. The device can detect SARS-CoV-2 S1 spike protein selectively, where the binding of the protein to the membrane-bound antibodies results in cellular bioelectric properties modification measured by Bioelectric Recognition Assay. The LOD is 1 fg/mL and the response time is about three minutes. In addition to that, the biosensor includes a portable read-out device that can be operated by a smartphone [226].

Moreover, nano-biosensors have shown an outstanding potential to contribute to the fight against COVID-19, providing holistic insights for developing ultrasensitive, cost-effective, and rapid detection devices for mass production [227]. Advanced materials are the basis of nano-enabled or integrated micro-and nano biosensing system technologies that can detect earlier the virus, and even show good binding properties allowing them to inactive or destroy the pathogen upon the application of an external stimulus [228].

Different research groups have developed carbon-based and graphene-based POC biosensors [214,223]. Graphene is foreseeing to have a leading role in the attempt of fighting against COVID-19. This low-cost material can be employed for virus detection since its sensitivity and selectivity can be enhanced by modifying its hybrid structure (e.g., antibody-conjugated graphene sheets) that allows tuning of its optical and electrical features. Some graphene-based sensors that can be explored for SARS-CoV-2 detection are photoluminescence, colorimetric, and SPR biosensors [229,230]. Seo et al. employed the material for the development of a field-effect transistor (FET)-based biosensor for detecting SARS-CoV-2 (Figure 5). In this case, graphene sheets from the FET were coated with a specific antibody against the virus spike protein, which was successfully detected at concentrations of 1 fg/mL in a phosphate-buffered saline medium. In addition, the device was able to detect the virus in clinical samples, exhibiting a LOD of 2.42 × 10^2^ copies/mL. The fabricated biosensor is considered as a promising immunological diagnostic alternative for the disease [231].

Additionally to FET, a review published by Cui et al. considers potential electrochemical biosensor and surface-enhanced Raman scattering (SERS)-based biosensor as other suitable options for diagnosis of COVID-19 [232]. Also, Murugan et al. designed two field-deployable/portable plasmonic fiber-optic absorbance biosensor (P-FAB) device for rapid detection of the virus’ N-protein directly from saliva. One of them was a labeled immunoassay, and the other one was label-free. Both bioanalytical approaches using the highly sensitive P-FAB platform can be considered as ideal alternatives for COVID-19 diagnosis within 15 min [233]. More recently, Zhu et al. reported another diagnosis approach based on the development of a multiplex reverse transcription loop-mediated isothermal amplification combined with NP-based lateral flow biosensor. The method allowed the multiplex detection of the open reading frame 1a/b (ORF1ab) and the N-protein within an hour, ensuring the sufficient sensitivity for the virus [234].

In another approach, Qiu et al. developed a dual-functional plasmonic biosensor that combines the plasmonic photothermal (PPT) effect and LSPR sensing transduction. The device is constituted by a two-dimensional gold nano island functionalized with complementary DNA receptors that can selectively detect specific sequences from SARS-CoV-2 through nucleic acid hybridization. In addition to that, PPT can increase the in situ hybridization temperature, which allows differentiating between two similar gene sequences. This biosensor showed high sensitivity with a lower LOD at 0.22 pM [235].

Although we have discussed several options for COVID-19 diagnosis, researchers are working on novel diagnostic techniques that combine different approaches based on nanotechnology and nanoscience, in order to obtain faster, reliable, and more accurate results that allow accelerating life-saving decisions, and isolation of positive patients in an early stage to down-regulate the virus spread [236,237].

## 5. Conclusions

In the last few decades, viral and bacterial pathogens have become a real menace to human safety. Their rapid identification must be considered as a priority task in order to prevent an outbreak that represents a high risk of disruption of the healthcare system, and a disastrous socio-economic impact. Scientists are performing intensive research for developing sensitive diagnostic techniques and effective therapeutics. There is no vaccine or pharmacological treatment for many viruses and bacteria and the development of a POC device for the rapid diagnosis of diseases such as COVID-19 allows accelerating life-saving decisions, and isolation of positive patients in an early stage. In this sense, biosensors and nano-biosensors are powerful measurement devices that can make the detection process of important clinical bacteria and virus to be easy, quick, and effective by sensing relevant parameters that can be related to infectious processes.

## Figures and Tables

**Figure 1 sensors-20-06926-f001:**
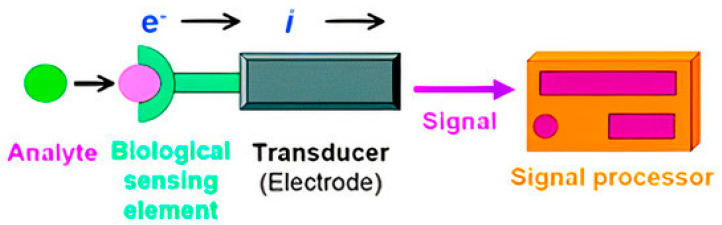
Biosensor’s basic design. Reprinted with permission from Huang, Y. et al. Disease-Related Detection with Electrochemical Biosensors: A Review. *Sensors* 17(10). Copyright (2017) MDPI [46].

**Figure 2 sensors-20-06926-f002:**
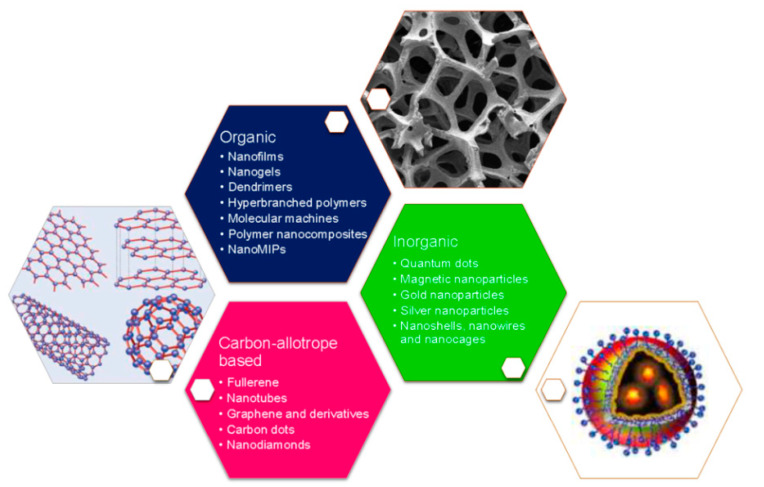
Different nanomaterials and nanostructures used for the development of nano-biosensors. Reprinted with permission from Pirzada, M. et al. Nanomaterials for Healthcare Biosensing Applications. *Sensors* 19(23): 5311. Copyright (2019) MDPI [95].

**Figure 3 sensors-20-06926-f003:**
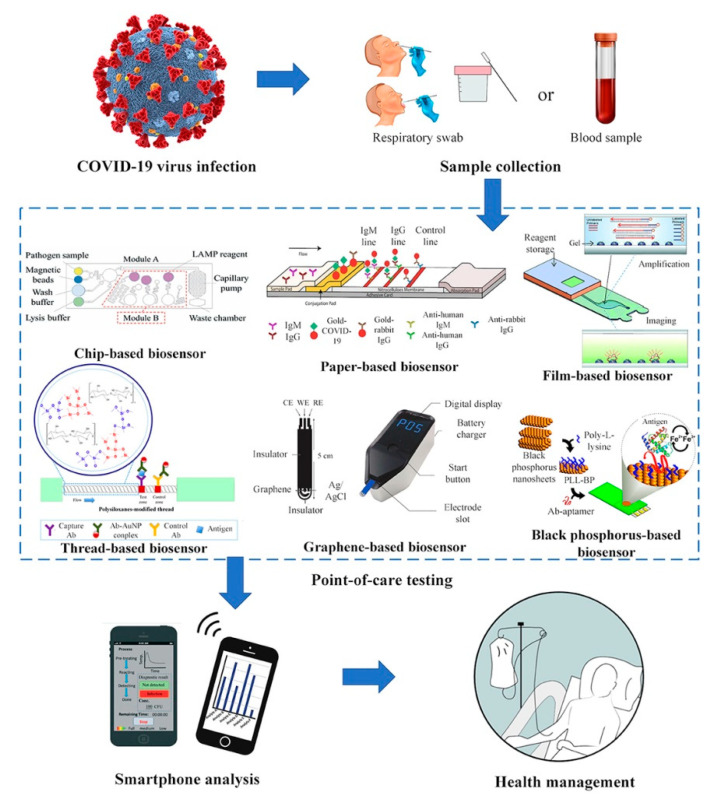
Point-of-care (POC) for COVID-19. Reprinted with permission from Choi, J. et al. Development of Point-of-Care Biosensors for COVID-19. *Front Chem* 8: 517. Copyright (2019) Frontiers in Chemistry [214].

**Figure 4 sensors-20-06926-f004:**
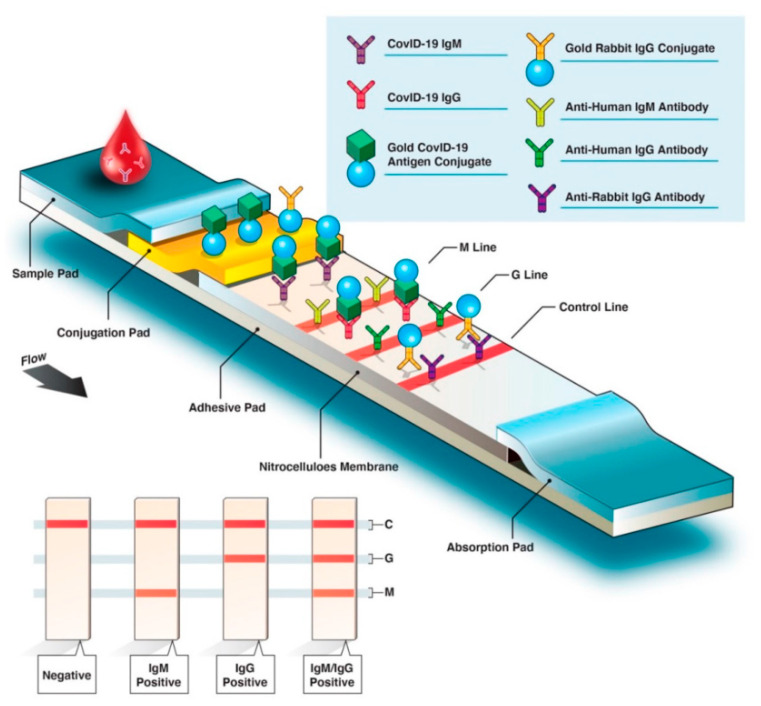
COVID-19 rapid serological IgM/IgG test. Reprinted with permission from Ghaffari, A. et al. COVID-19 Serological Test: How Well Do They Actually Perform? *Diagnostics* 10(7): 453. Copyright (2020) MDPI [221].

**Figure 5 sensors-20-06926-f005:**
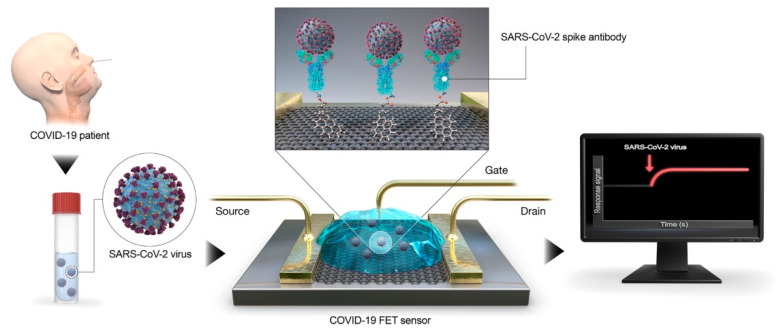
Schematic diagram of COVID-19 FET sensor operation procedure. Reprinted with permission from Seo, G. et al. Rapid Detection of COVID-19 Causative Virus (SARS-CoV-2) in Human Nasopharyngeal Swab Specimens Using Field-Effect Transistor-Based Biosensor. *ACS Nano* 14(4): 5135–5142. Copyright (2020) ACS [231].

**Table 1 sensors-20-06926-t001:** Developed biosensors for detecting bacterial and viral pathogens in the human body.

Device	Target Pathogen	LOD	Response Time	Reference
Long-period fiber grating using bacteriophage T4 covalently immobilized on optical fiber surface.	*E. coli*	10^3^ CFU/mL	20 min	[142]
Label free polyaniline based impedimetric.	*E. coli* O157:H7	10^2^ CFU/mL	-	[143]
Electrochemical biosensor using antibody-modified NPs (polymer-coated magnetic NPs and carbohydrate-capped AuNPs).	*E. coli* O157:H7	10^1^ CFU/mL	45 min	[144]
Graphene-based potentiometric.	*S. aureus*	1 CFU/mL	10–15 min	[145]
Aptamer based biosensor and dual florescence resonance energy transfer from QDs to carbon NPs.	*Vibrio parahaemolyticus* and *Salmonella typhimurium*	25 CFU/mL and 35 CFU/mL, respectively	80 min	[116]
Impedimetric biosensor based on site specifically attached engineered antimicrobial peptides.	*Pseudomona aeruginosa*	10^2^ CFU/mL	30 min	[146]
Electrochemical DNA biosensor based on flower-like ZnO nanostructures.	*Neisseria meningitides*	5 ng/μL	-	[147]
Graphene-enabled biosensor with a highly specific immobilized monoclonal antibody.	Zika virus	0.45 nM	4–8 min	[148]
Giant magnetoresistance biosensor.	Influenza A virus	1.5 × 10^2^ TCID_50_/mL	-	[149]
Electrochemical biosensor based on DNA hybridization.	Hepatitis A virus	6.94 fg/μL	15 min	[150]
Impedimetric electrochemical DNA biosensor for label free detection.	Zika virus	25 nM	1.5 h	[151]
Two-dimensional molybdenum disulphide nanosheets based disposable biosensor.	Chikungunya virus	3.4 nM	3 h	[152]
Electrochemical DNA biosensor using gold nanorods.	Hepatitis B virus	2.0 × 10^−12^ mol/L	5 h	[153]
Intensity-modulated surface plasmon resonance (IM-SPR) biosensor	Avian influenza A H7N9 virus	144 copies/mL	10 min	[154]
Silicon nanowire biosensor.	Dengue virus	2.0 fM	-	[155]

AuNPs: gold nanoparticles; *E. coli*: *Escherichia coli*; IM-SPR: Intensity-modulated Surface Plasmon Resonance; LOD: limit of detection; NPs: nanoparticles; QDs: quantum dots; *S. aureus*: *Staphylococcus aureus*; SPR: Surface Plasmon Resonance.

**Table 2 sensors-20-06926-t002:** FDA commercially authorized biosensors for SARS-CoV-2 detection [220].

Manufacturer	Device	Target	Clinical Combined Specificity	Clinical Combined Sensitivity
Abbott	SARS-CoV-2 IgG chemilumininescent microparticle immunoassay (CMIA)	Nucleocapsid	99.9%	100%
Access Bio, Inc.	CareStart COVID-19 IgM/IgG	Spike and Nucleocapsid	98.9%	98.4%
Beijing Wantai Biological Pharmacy Enterprise Co. Ltd.	Wantai SARS-CoV-2 Ab rapid test	Spike	98.8%	100%
Biohit Healthcare (Hefei)	Biohit SARS-CoV-2 IgM/IgG antibody test kit	Nucleocapsid	95.0%	96.7%
Cellex	Cellex Qsars-CoV-2 IgG/IgM rapid test lateral flow immunoassay	Spike and nucleocapsid	96.0%	93.8%
DiaSorin	LIAISON SARS-CoV-2 S1/S2 IgG CMIA	Spike	99.3%	97.6%
Hangzhou Biotest Biotech	COVID-19 IgG/IgM rapid test cassette	Spike	100%	100%
Hangzhou Laihe Biotech	LYHER novel coronavirus (2019-nCoV) IgM/IgG antibody combo test kit (colloidal gold)	Spike	98.8%	100%
Healgen	COVID-19 IgG/IgM rapid test cassette	Spike	97.5%	100%
Megna Health, Inc.	Rapid COVID-19 IgM/IgG combo test kit	Nucleocapsid	95%	100%
Salofa Oy	Siena-Clarity COVIBLOCK COVID-19 IgG/IgM Rapid test cassette	Spike	98.8%	93.3%
Xiamen Biotime Biotechnology Co., Ltd.	BIOTIME SARS-CoV-2 IgG/IgM rapid qualitative test	Spike	96.2%	100%

CMIA: chemilumininescent microparticle immunoassay; COVID-19: coronavirus disease 2019.

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
