# Peer review of "Biosensors for the Detection of Bacterial and Viral Clinical Pathogens"

_sensors, 2020, doi:10.3390/s20236926_

Round 1

Reviewer 1 Report

Please, read the PRISMA guidelines checklist. The authors should present their review process with: Database searched (e.g. Web of Science, Science direct), terms used in the search, data range (year range for the search), number of articles retrieved, number of articles filtered (How many articles have been removed from the first withdrawn) and filter (The reason for the removal of the first withdrawn articles).

L 237. “NPs” Titles and subtitles are better with the full name and not with abbreviations

L 267. “GQDs” Same as above

L 436. 5. COVID-19 pandemic SHOULD BE 4.2.1 COVID-19 pandemic

Finally, in my personal opinion, despite "the current appeal", the exclusion of  "COVID 19" from title would be fair to the current proposal. Whereas, this term can be kept in key words.

Reviewer 2 Report

Review article titled "Biosensors for the Detection of Bacterial and Viral 2 Clinical Pathogens and COVID-19 Diagnosis"is dedicated for topic of biosensors for bacterial and viral detection, and the nanotechnological features that can be applied for a faster  COVID-19 diagnosis at the point-of-care. 
This review article is interesting and could be useful for a broad spectrum of reeder after major revision.
1. There are many sencenses where it not clear what authors want to say as in a row 85:In order to obtain a high-quality biosensor, researchers must identify the requirements to obtain a fully functional device.
And in a row 90:These devices can monitor several parameters like disease-causing bacteria, and several body fluids such as saliva, blood, or urine.
Disease-causing bacteria, several body fluids such as saliva, blood, or urine are not the monitor parameters in biosensors.

2. Authors are reviewing many other biosensing methods dedicated for bacterial and viral detection but just a little information about optical biosensors are presented. Such methods as surface plasmon resonance (SPR), spectroscopic ellipsometry in total internal reflection mode (TIRE) should be also reviewed and discussed as these methods are very sensitive, non destructive for detection of bio objects as antigens and antibodies or receptors ligands, also cells, they do not require any labeling and provide rapid detection in real time. The information about these methods should be expanded.

3. The title of article "Biosensors for the Detection of Bacterial and Viral 2 Clinical Pathogens and COVID-19 Diagnosis" is inappropriate as detection of specific antigens or antibodies for SARS COV2 is not the diagnosis of COVID-19 illness. Please suggest more correct title for the article, because now the title do not represent the information written in the text  

Round 2

Reviewer 2 Report

The article can be accepted in present form